# Pre-Pregnancy BMI Influences the Association of Dietary Quality and Gestational Weight Gain: The SECOST Study

**DOI:** 10.3390/ijerph16193735

**Published:** 2019-10-04

**Authors:** Heng Yaw Yong, Zalilah Mohd Shariff, Barakatun Nisak Mohd Yusof, Zulida Rejali, Yvonne Yee Siang Tee, Jacques Bindels, Eline M. van der Beek

**Affiliations:** 1Department of Nutrition and Dietetics, Faculty of Medicine and Health Sciences, Universiti Putra Malaysia, Seri Kembangan 43400, Malaysia; yong_hy@upm.edu.my (H.Y.Y.); bnisak@upm.edu.my (B.N.M.Y.); 2Department of Obstetrics and Gynecology, Faculty of Medicine and Health Sciences, Universiti Putra Malaysia, Seri Kembangan 43400, Malaysia; zulida@upm.edu.my; 3Danone Dumex (M) Sdn Bhd. Lot 759 (B3), Nilai Industrial Estate, Nilai 71800, Malaysia; yvonneyeesiang.tee@danone.com; 4Danone Nutricia Research, Uppsalalaan 12, 3584 CT Utrecht, The Netherlands; jacques.bindels@danone.com (J.B.); eline.vanderbeek@danone.com (E.M.v.d.B.); 5Department of Pediatrics, University Medical Centre Groningen, University of Groningen, 9713 GZ Groningen, The Netherlands

**Keywords:** diet quality, healthy eating index (HEI), gestational weight gain (GWG), pre-pregnancy BMI

## Abstract

Poor diet quality in pregnancy could impact gestational weight gain (GWG) and consequently fetal growth and development. But today there is limited data available on gestational diet quality. This study investigated the association between diet quality in each pregnancy trimester and GWG in Malaysian women. Diet quality was assessed using the modified Healthy Eating Index for Malaysians (HEI). Total GWG was defined as the difference between measured weight at last prenatal visit and pre-pregnancy weight. About one-fourth of women (23.3%) had excessive total GWG. There were significant differences in the HEI component score across trimesters, except for fruits. Overall, overweight/obese women had lower total HEI score (51.49–55.40) during pregnancy compared to non-overweight/obese women (53.38–56.50). For non-overweight/obese women, higher total HEI scores in the second and third trimesters were significantly associated with lower risk of inadequate GWG (aOR = 0.97, 95% CI = 0.95–0.99, *p* = 0.01) and higher risk of excessive GWG (aOR = 1.04, 95% CI = 1.01–1.07, *p* = 0.03), respectively. Overweight/obese women with higher total HEI scores in the second (aOR = 1.04, 95% CI = 1.01–1.07, *p* = 0.02) and third trimester (aOR = 1.04, 95% CI = 1.01–1.08, *p* = 0.02) were significantly at higher risk for excessive GWG. Pregnant women had relatively low diet quality throughout pregnancy. Diet quality and GWG association differed according to pre-pregnancy BMI with excessive GWG more likely to be associated with higher total HEI scores in the third trimester.

## 1. Introduction

To date, there is no Asian-specific gestational weight gain (GWG) guideline. With the exception of Japan, most Asian countries use the Institute of Medicine (IOM) GWG guideline [1]. In 2009, the US Institute of Medicine (IOM) established updated guidelines for weight gain during pregnancy, recommending that GWG should be based on pre-pregnancy BMI. Pregnant women with high pre-pregnancy BMI should gain less weight in pregnancy than those with a lower pre-pregnancy BMI. Nevertheless, most studies showed that overweight or obese women tend to gain above the recommended range, while underweight women were more likely to have inadequate GWG [2,3]. Previous studies in Malaysia have shown that more than one-third of pregnant women were overweight or obese, and more than half of these women had excessive GWG [4,5]. It is well-documented that excessive GWG is associated with increased risk of adverse pregnancy and birth outcomes [6].

Diet quality indices are increasingly being used to determine associations between dietary intake with nutritional status and health outcomes [7]. Such indices measure how well the diets conform to the recommendations based on national dietary guidelines [8]. Generally, pregnant women who consume sufficient amounts of food tend to have better nutritional status and pregnancy outcomes [9]. Nonetheless, it is also crucial for pregnant women to consider the overall quality of their diet. As there are variations in dietary guidelines across countries and cultural differences in population diets, several versions of diet quality indices for pregnant women have been developed [10,11,12]. Most studies on diet quality of pregnant women described the diet quality in specific trimester of pregnancy (early pregnancy or 26–28th weeks) [13,14,15] and determined its associated factors [14,16,17]. However, limited studies examined diet quality for each trimester [17,18] and the possible association between diet quality and GWG [15,19]. As diets of women tend to change over the course of pregnancy, it is important to assess the diet quality of women in each trimester separately, as the they might have different implications for pregnancy outcomes.

Both excessive and inadequate GWG can have negative consequences on pregnancy and birth outcomes [20]. Mothers with inadequate GWG have higher risk for seizure, longer hospital stay, miscarriage, and delivery of small-for-gestational (SGA) infants [21,22,23]. Excessive GWG is associated with increased risk for caesarean delivery, preeclampsia, and large-for-gestational-age (LGA) births [21,24,25,26]. Both situations may also have long term implications for growth and development and the risk for disease later in life [27,28]. Thus, it is crucial for child-bearing age women to adopt and maintain healthy eating habits to ensure healthy pre-pregnancy BMI and achieve optimal GWG during pregnancy. Improvements in women’s health and related behaviors are likely to have benefits not only for their own health but also for the health of their offspring. Unlike many other risk factors that vary by socioeconomic position, dietary intake is modifiable. This study described the diet quality of Malaysian women during pregnancy that reflects the national dietary guidelines, and its association with total GWG, observing differential effects of diet quality in the first, second and third trimester on total GWG.

## 2. Materials and Methods 

### 2.1. Study Design and Location

SECOST (Seremban Cohort Study) is a prospective study in which pregnant women were followed-up through 1 year postpartum, and their infants were followed-up every six months until two years of age. Women in the first trimester (10–13th weeks of gestation) of pregnancy were recruited from three Maternal and Child Health (MCH) clinics in Seremban District, Negeri Sembilan, Malaysia. 

### 2.2. Measurements

All women were interviewed by trained enumerators using a pre-tested questionnaire. Details of the instrument have been published elsewhere [29]. The instruments were pre-tested prior to data collection as to ensure the appropriateness, clarify and interpretation of the instruments. Socio-demographic information obtained included current age, education level, ethnicity, occupation status, monthly household income, and household size. Obstetrical information (e.g., gravidity and parity) were obtained from medical records. 

### 2.3. Dietary Assessment

A one-day, 24-h dietary recall was used to obtain food intakes of respondents at each trimester. Women were required to recall all food and beverages consumed in the past 24 h. The recall form consisted of types of food and beverages, time of eating or drinking, food ingredients, preparation methods, and quantity of foods and beverages consumed. Standard calibrated household measuring cups, glasses, bowls, and spoons were used to assist the respondents in recalling the portion size of food and beverages consumed. Dietary data were analyzed using Nutritionist Pro Diet Analysis software: Version 1.5 (Axxya Systems, CA, USA) [30]. Food intake data were also presented as number of servings consumed from each food group based on the Malaysian Food Guide Pyramid [31]. 

### 2.4. Healthy Eating Index (HEI) 

Diet quality of pregnant women was assessed using the modified Healthy Eating Index for Malaysians (HEI) (Appendix A). The HEI comprised nine components, each representing different aspects of a healthful diet. Components 1–7 measured the person’s degree of compliance with the seven major food groups: Cereals and grains, vegetables, fruits, milk and milk products, poultry, meat and egg, fish and seafood, and legumes, recommended by Malaysian Dietary Guidelines 2010 for Malaysian (MDG). Component 8–9 measured the compliance with the recommendation of the percentage of energy from fat, and total sodium intake by MDG [31]. Each component has a maximum score of 10 for full compliance and a minimum score of 0 for lack of compliance. The score for each component was calculated using the formula: (Actual serving consumed based on respondent’s diet recall/ recommended serving size based on MDG) and multiplied by 10. If an individual consumed less than the recommended amount of servings, the score was calculated with the following formula: 10 × (the consumed amount of servings)/ (the lower limit of the recommended serving). If an individual consumed more than the recommended amount of servings, the score was calculated with the following formula: 10–10 × [(the consumed servings) − (the upper limit of the recommended servings)]/(the upper limit of the recommended serving). Each score was rounded off to the nearest whole number. When this calculation produced a negative score because of excess servings, the score was converted to 0. The score was calculated proportionately for the in-between responses [8,32]. Total HEI score was calculated by summing up the score of each component. The possible score for total HEI ranges from 0 to 100. A higher score indicates an intake close to the recommended range, while a lower score reflects less compliance with recommended intakes. 

### 2.5. Anthropometric Measurements 

Maternal height was measured at study enrolment, while weight was measured at each study visits using a standard instrument (SECA digital weighing scale and SECA body meter) and standard procedures. Women were requested to recall pre-pregnancy body weight (current pregnancy). Pre-pregnancy BMI was calculated from height and weight and categorized using World Health Organization (WHO)’s cut-off points respectively: Underweight (< 18.5 kg/m^2^), normal weight (18.5–24.9 kg/m^2^), overweight (25.0–29.9 kg/m^2^), and obese (≥30.0 kg/m^2^) [33]. Weight in the first, second and third trimester was measured as the closest measurement to 12th weeks of gestation (10–13th weeks), the closest measurement to 26 weeks of gestation (range 24–32nd weeks), and the closest measurement to 38 weeks of gestation (range 34–38th weeks). Total GWG was defined as the difference between the measured weight at last prenatal visit and the pre-pregnancy weight. Rate of GWG in the second trimester and third trimester was defined as the average weekly weight gain in that trimester and was then categorized as inadequate, adequate, or excessive for each category of pre-pregnancy BMI [34]. Total GWG in relation to pre-pregnancy BMI was then classified as gaining below (inadequate GWG), within (adequate GWG), or above (excessive GWG) the recommendation of IOM [34]. 

### 2.6. Other Variables

The Pregnancy Physical Activity Questionnaire (PPAQ) was used to determine the physical activity level of pregnant women [35]. PPAQ consisted of items on the frequency and intensity of physical activity (PA), frequency of vigorous PA, hours spent on vigorous PA, the average duration of a PA session. Total activity was calculated as the sum of all intensity activities and type scores. A metabolic equivalent (MET) hours per week was calculated by multiplying the duration of time spent in each activity with an established MET value.

### 2.7. Statistical Analysis 

All statistical analyses were performed using IBM SPSS Statistics for Windows, Version 25.0 (IBM, New York, USA) [36]. Continuous variables were expressed as the means and standard deviations, while categorical variables as absolute frequencies and percentages. Multinomial logistic regression was performed to determine the associations between diet quality in each trimester and total GWG adjusted for covariates. Covariates (continuous variables) included were age, years of education, parity, physical activity level, and pre-pregnancy BMI. Given the possibility of an interaction effect between age, years of education, parity, physical activity, and pre-pregnancy BMI with diet quality, models incorporating interaction terms were also performed. Analysis of covariance (ANCOVA) with Bonferroni correction for multiple comparisons was used to determine the associations between diet quality and total GWG stratified by pre-pregnancy BMI categories (non-overweight/obese vs. overweight/obese). Statistical significance was set at *p* < 0.05.

## 3. Results

Table 1 presents the characteristics of women. The mean age of the women was 30.16 ± 4.51 years, with 52.3% aged over 30 years. Most of the women were Malay (89.0%), had secondary or lower education (46.0%), were employed (69.2%), and had low monthly household income (63.5%). The mean gravidity and parity of women were 2.46 ± 1.48, and 1.22 ± 1.29, respectively. About 7.5% of women had a medical history of GDM and more than one-fourth had a family history of diabetes mellitus (26.9%). The mean pre-pregnancy weight and pre-pregnancy BMI were 59.11 ± 13.57 kg and 24.10 ± 5.06 kg/m^2^, respectively. More than half (53.1%) had normal pre-pregnancy BMI (18.50–24.99 kg/m^2^), while about 22.3% and 14.4% were categorized as overweight and obese respectively. The mean total energy expenditure in the second and third trimester was 264.58 ± 118.06 Mets hours/week and 249.56 ± 107.36 Mets hours/week, respectively. This finding was slightly higher than those reported in previous studies [37,38,39].

The overall mean rate of GWG at third trimester (0.39 ± 0.01 kg/week) was slightly higher than the second trimester (0.37 ± 0.01 kg/week) (Table 2). The mean total GWG was 11.06 ± 0.23 kg. Most of the underweight (89.8%) and normal weight (95.5%) women had inadequate to adequate total GWG. About 43.9% of overweight women had adequate total GWG. Among obese women, more than half (52.2%) had excessive total GWG.

The mean total HEI scores and HEI component score for the first, second and third trimester are presented in Table 3. The mean total HEI score in the first (52.73 ± 0.52) and the third trimester (52.76 ± 0.52) were almost similar. The HEI score has improved at the second trimester with the HEI score of 57.10 ± 0.52. There were significant differences in HEI component scores across trimesters, except for fruits. Women had higher HEI score for cereals and grains (7.53–8.54), poultry, meat, and egg (7.52–8.55) and sodium (7.04–8.61), but lower HEI score for legumes (1.04–3.14) and milk and milk products (1.96–3.82).

Table 4 shows the adjusted odds ratio for associations between diet quality and total GWG. Women with higher total HEI score in the second trimester had significantly lower risk of inadequate GWG (aOR = 0.98, 95% CI = 0.96–0.98) after adjusted for covariates. Meanwhile, women with higher total HEI score in the third trimester were at significantly higher risk of excessive GWG (aOR = 1.04, 95% CI= 1.04, 95% CI = 1.01–1.06) after adjusted for covariates. The likelihood ratio test revealed that only pre-pregnancy BMI showed significant interactions with total HEI score in second (χ^2^ = 33.38, *p* < 0.01) and third trimester (χ^2^ = 42.08, *p* < 0.01) to the risk of excessive GWG. The mean total HEI score for non-overweight/obese women at the first, second, and third trimester were 53.45 ± 0.68, 56.50 ± 0.64 and 53.38 ± 0.64, respectively. Overall, a significantly lower total HEI score across trimesters of pregnancy was found among overweight/obese women with the mean total HEI score of 51.48 ± 0.84, 55.40 ± 0.89 and 51.69 ± 0.81 for the first, second and third trimester (F = 15.05, *p* = 0.03). 

The associations between total HEI score in each trimester and total GWG by pre-pregnancy BMI are shown in Table 5. Non-overweight/obese women with a higher total HEI score in the second trimester had lower risk of inadequate GWG (Aor = 0.97, 95% CI= 0.95–0.99) but a higher total HEI score in the third trimester was significantly associated with higher risk of excessive GWG in this group (aOR = 1.04, 95% CI = 1.01–1.07). For overweight/obese women, higher total HEI scores in both second (aOR = 1.01, 95% CI = 1.01–1.02) and third trimesters (aOR = 1.04, 95% CI = 1.01–1.04) were significantly associated with higher risk for GDM.

## 4. Discussion

The present study showed that non-overweight/obese women with higher total HEI scores in the second trimester showed a lower risk for inadequate GWG, yet higher total HEI scores in the third trimester were significantly associated with a higher risk of excessive GWG. Further analysis to determine the components of HEI that were associated with GWG among this group of non-overweight/obese women showed that in particular higher intakes of cereals and grains in the second trimester was associated with a significantly lower risk of inadequate GWG (aOR = 0.71, 95% CI = 0.55–0.92, *p* < 0.05). Although carbohydrates are an important component of a healthy diet during pregnancy [40], its association with GWG has been inconsistent [41,42] which could be related to the type and quality of carbohydrate. Carbohydrate sources with lower glycemic index (GI) such as whole grains, contribute to lower energy density, increased satiety, and subsequently adequate GWG; conversely, carbohydrate sources with higher GI such as refined grains (e.g. desserts, and sweet snacks) tend to be high in energy density and could contribute to higher GWG [43]. The cereals and grains commonly consumed by women in this study included rice, noodles and pasta, bread, cereal, and cereals products, which are considered as good carbohydrate sources. Nevertheless, further studies are needed to confirm the reported associations and to determine which food types in cereals and grains group contribute to (adequate) GWG.

While overweight/obese women with higher total HEI scores in the second and third trimester had a higher risk for excessive GWG, non-overweight/obese women with higher total HEI scores in the third trimester were also more likely to have excessive GWG. Analyses of HEI components revealed that overweight/obese women with higher intakes of fruits in the second (aOR = 1.19, 95% CI = 1.07–1.32, *p* < 0.05) and the third trimester (aOR = 1.13, 95% CI = 1.03–1.24, *p* < 0.05), as well as higher intakes of milk and milk products in the third trimester (aOR = 1.03, 95% CI = 1.01–1.31, *p* < 0.05) had significant risk for excessive GWG. Similarly, normal weight with higher intakes of fruits (aOR = 1.03, 95% CI = 1.01–1.15, *p* < 0.05) and milk and dairy products (aOR = 1.06, 95% CI = 1.02–1.19, *p* < 0.05) in the third trimester were more likely to have excessive GWG. In this sample of pregnant women, frequently consumed fruits were bananas, grapes, mangos, dates, raisins, and durian. Milk and milk products include milk, ice-cream, and cheese. Although women in this study had low HEI component scores for fruits, vegetables and milk/milk products, the nutrient content and methods of food preparation could contribute to the energy density of the foods. The commonly consumed fruits are not only relatively high in sugar, particularly if consumed in large amount, but also are frequently consumed as juices/shakes/blended ice/traditional sweet desserts with added sugar. Similarly, milk may be consumed as plain but could also be a flavoured milk or may be added to beverages such as sugar added tea and coffee, milk shakes and malted drinks. Although not all types of milk products have high energy and fat/sugar contents, low fat, and low-sugar ice-creams and low-fat cheeses are not commonly available in the market. Thus, choosing high energy dense foods (high sugar and/or fat) may result in an increased daily total intake of calories which could contribute to higher weight gain.

The mean total HEI score (52.76–56.10) for pregnant women in this study was substantially lower than those reported in Western countries (62.9–70.2) [16,17,44,45] but almost parallel to those in other Asian countries, such as Singapore (52.4) [13] and Indonesia (58.9) [15]. Cross-study variations in HEI scores could be explained by the differences in study design, measurement of dietary intake (e.g., diet recall, diet history, food frequency questionnaire), socio-demographic background (e.g., age, ethnicity, and nutritional knowledge) and food environment (e.g., accessibility to fast food, convenience store and food court setting). Based on the analyses of HEI components, women in the present study had low HEI component score for legumes and milk and milk products, with the average score ranging from 1.04 to 3.83. This is consistent with the findings of a review and meta-analysis on energy and macronutrient intakes of Malaysian adults which showed that most Malaysians did not meet the recommended servings for other protein sources specifically legumes, nuts, milk and milk products [46]. 

A recent study by Savard et al. (2019) showed that there was no significant variation in overall diet quality across all trimesters of pregnancy among Canadian pregnant women. For HEI component analysis, intake of fruits and vegetables decreased significantly throughout pregnancy, but intake of milk and milk products increased significantly across trimester [17]. In contrast, Moran et al. (2013) reported an overall decrease in maternal diet quality during pregnancy in overweight/obese women [18]. However, the present study highlights that there was significant variation in the total HEI score across all trimesters of pregnancy, whereby total HEI score increased significantly from the first to the second trimester and then decreased in the third trimester to a level that was similar in the first trimester. Given that this change occurred in the context of a relatively low total HEI score, this finding should be interpreted with caution. A possible explanation for the increase in the total HEI scores from the first to the second trimester could be due to the resolution of nausea or vomiting after the first trimester or the positive dietary changes after receiving nutrition advice during early pregnancy [18]. Although pregnancy is known to be a period during which pregnant women are motivated to adopt healthy behaviors, it is also possible that motivation decreases as pregnancy progresses, making it difficult for women to maintain high quality of their diets.

At present, there is no standard indicator or measure of diet quality. The HEI was established by the United States Department of Agriculture (USDA) in 1995 to measure how well the diet conforms to the national dietary guidelines [8]. It has been widely used as a measure of diet quality in all life stages [47,48], including pregnancy and in relation to various health or disease outcomes [49,50,51]. As dietary guidelines vary by countries and cultures, several versions of HEI for pregnant women have been developed [10,11,12]. The HEI-1995, HEI-2005, and the Alternate Healthy Eating Index for Pregnancy (AHEI-P) are commonly used diet quality indexes for the US pregnant women [12,16,52,53,54]. Asian countries, such as China [55], Singapore [13], and Indonesia [15] have adapted the HEI and AHEI-P to the local dietary guidelines. These measures of HEI were also associated with sociodemographic characteristics [53,54,56], pre-pregnancy weight status [12,16], and birth outcomes [46,57,58]. The HEI for Malaysians used in this study showed an association with GWG, thus, giving support to HEI as a good indicator of diet quality in this study population.

This study has several limitations. Respondents were not representative of the general population of pregnant women in Malaysia. Most women were Malay, had secondary education and lower, and were of low- and middle-income households. Besides recall bias due to self-report, the use of one 24-h dietary recall to assess diet quality of pregnant women might not represent the usual intake of pregnant women. To prevent under-reporting, albums of foods and beverages and household measurements were used to assist the respondents’ recall of dietary intake. The present study was limited to describing diet quality and its association to GWG and did not explore factors (e.g. socioeconomic position, food environment, motivation to change) influencing eating behavior that may shape diet quality during pregnancy. The impact of diet quality could very well extend beyond GWG to birth outcomes (e.g. macrosomia, low birthweight and pre-term delivery), but such outcomes were not investigated in the present study. Regardless of these limitations, this study was able to provide valuable insights into diet quality and the relationship with weigh gain during pregnancy. These results could inform the development of recommendations and prevention strategies to improve pregnancy outcome.

## 5. Conclusions

Overall, pregnant women in the current study had relatively low HEI scores and the total HEI score varied throughout pregnancy. Women who were overweight and obese had poorer HEI score during pregnancy compared to non-overweight/obese women. Diet quality was significantly associated with GWG and this association differed significantly between non-overweight/obese and overweight/obese women. Women with higher total HEI score in third trimester were at higher risk for excessive GWG, regardless of pre-pregnancy BMI. Assessment of diet quality and its association to GWG is needed to develop tailored interventions for pregnant women that ensure adequate diet quality and gestational weight gain through healthy food choices and micronutrient supplementation.

## Figures and Tables

**Table 1 ijerph-16-03735-t001:** Characteristics of women (*n* = 480).

Variables	*n* (%)	Mean ± SD
Age (years)		30.16 ± 4.51
≤30	229 (47.7)	
>30	251 (52.3)	
Ethnicity		
Malay	427 (89.0)	
Non-Malay	53 (11.0)	
Education level (years)		12.95 ± 2.41
Secondary and lower	221 (46.0)	
STPM/ matric/ diploma/ certificate	157 (32.7)	
Tertiary and above	102 (21.3)	
Occupation status		
Unemployed	148 (30.8)	
Employed	332 (69.2)	
Monthly household income (RM) ^1^		3698.30 ± 2034.20
Low (<3860)	305 (63.5)	
Middle (3860–8319)	161 (33.5)	
High (≥8320)	14 (2.9)	
Household size		3.78 ± 1.63
≤2	116 (24.2)	
3–4	240 (50.0)	
≥5	124 (25.8)	
Gravidity		2.46 ± 1.48
1	155 (32.3)	
2	132 (27.5)	
≥3	193 (40.2)	
Parity		1.22 ± 1.29
0	176 (36.7)	
1–2	229 (47.7)	
≥3	75 (15.6)	
Medical history of GDM		
No	444 (92.5)	
Yes	36 (7.5)	
Family history of diabetes mellitus		
No	351 (73.1)	
Yes	129 (26.9)	
Height (m)		1.56 ± 0.06
<1.55	176 (36.7)	
1.55–1.58	144 (30.0)	
≥1.59	160 (33.3)	
Pre-pregnancy weight (kg)		59.11 ± 13.57
Pre-pregnancy BMI (kg/m^2^)		24.10 ± 5.06
Underweight (<18.5)	49 (10.2)	
Normal (18.5–24.9)	255 (53.1)	
Overweight (25.0–29.9)	107 (22.3)	
Obese (≥30.0)	69 (14.4)	
Physical activity (MET-hours/week)		
2nd trimester		264.58 ± 118.06
3rd trimester		249.56 ± 107.36

^1^ 10th Malaysia Plan, 1 USD = RM 4.18.

**Table 2 ijerph-16-03735-t002:** Gestational weight gain (GWG) of women by pre-pregnancy BMI (*n* = 480).

Pre-pregnancy BMI (kg/m^2^) ^1^	GWG in kg	Rates of GWG in kg/week ^2,3^		Total Weight Gain in kg ^2,3^			
	1st Trimester	2nd Trimester	3rd Trimester				
	Median (IQR)	Mean ± SD		Mean ± SD	I	A	E
Underweight (*n* = 49)	3.00 (4.25)	0.46 ± 0.02	0.40 ± 0.03	12.93 ± 0.54	22 (44.9)	22 (44.9)	5 (10.2)
Normal weight (*n* = 255)	2.00 (3.50)	0.41 ± 0.01	0.43 ± 0.02	11.90 ± 0.29	115 (51.1)	100 (44.4)	40 (4.5)
Overweight (*n* = 107)	2.00 (4.00)	0.31 ± 0.02	0.32 ± 0.03	9.49 ± 0.42	29 (27.1)	47 (43.9)	31 (29.0)
Obese (*n* = 69)	2.00 (5.05)	0.28 ± 0.03	0.36 ± 0.04	9.03 ± 0.73	15 (21.7)	18 (26.1)	36 (52.2)
Total (*n* = 480)	2.00 (4.00)	0.37 ± 0.01	0.39 ± 0.01	11.06 ± 0.23	181 (37.7)	187 (39.0)	112 (23.2)

Rate of weight gain was defined as average weekly weight gain in that particular trimester of pregnancy. Total weight gain was defined as the difference between the weight at last prenatal visit and pre-pregnancy weight. ^1^ Based on IOM recommended range for weight gain during pregnancy (2009). ^2^ Underweight (<18.5); normal weight (18.5–24.9); overweight (25.0–29.9); obese (≥30.0). ^3^ I = inadequate; A = adequate; E = excessive.

**Table 3 ijerph-16-03735-t003:** Component score and total score for Healthy Eating Index for Malaysians (HEI) by trimester of pregnancy.

Variables	Possible Range of Score	1st Trimester	2nd Trimester	3rd Trimester	F	*p*-Value	*p*-Value for Trend
		Mean ± SE					
HEI component							
Cereals and grains	0 to 10	8.54 ± 0.09 ^a,b^	9.37 ± 0.09 ^a,c^	7.53 ± 0.09 ^b,c^	97.63	0.001 *	0.001 *
Vegetables	0 to 10	3.03 ± 0.14 ^a,b^	4.20 ± 0.14 ^a^	4.14 ± 0.14 ^b^	21.75	0.001 *	0.001 *
Fruits	0 to 10	3.43 ± 0.18	4.00 ± 0.18	3.67 ± 0.18	2.55	0.07	0.35
Poultry, meat and egg	0 to 10	8.48 ± 0.14 ^b^	8.55 ± 0.14 ^c^	7.52 ± 0.14 ^b,c^	16.06	0.001 *	0.001 *
Fish and seafood	0 to 10	5.83 ± 0.20 ^a^	6.84 ± 0.20 ^a,c^	5.93 ± 0.20 ^c^	7.60	0.001 *	0.72
Legumes	0 to 10	2.20 ± 0.17 ^a,b^	1.04 ± 0.16 ^a,c^	3.14 ± 0.17 ^b,c^	40.96	0.001 *	0.001 *
Milk and milk products	0 to 10	1.96 ± 0.15 ^a,b^	3.82 ± 0.15 ^a,c^	2.73 ± 0.15 ^b,c^	38.78	0.001 *	0.001 *
% of energy from total fat	0 to 10	5.37 ± 0.21 ^b^	5.59 ± 0.21 ^b^	4.68 ± 0.21 ^b,c^	5.23	0.01 *	0.02 *
Sodium	0 to 10	8.61 ± 0.14 ^a,b^	7.04 ± 0.14 ^a,c^	7.77 ± 0.14 ^b,c^	31.57	0.001 *	0.001 *
Total HEI	0 to 100	52.73 ± 0.52 ^a^	56.10 ± 0.52 ^a,c^	52.76 ± 0.52 ^c^	14.01	0.001 *	0.001 *

Adjusted by age, years of education, parity, total GWG, physical activity level (MET-hours/week) and pre-pregnancy BMI. Means with similar superscripts in the same row indicate a significant difference (*p* < 0.05): ^a^ first trimester vs. second trimester; ^b^ first trimester vs. third trimester; ^c^ second trimester vs. third trimester. * *p* < 0.05.

**Table 4 ijerph-16-03735-t004:** Adjusted OR for associations between total HEI score in each trimester and GWG.

Diet Quality	Inadequate GWG		Excessive GWG	
	aOR (95% CI)	*p*-Value	aOR (95% CI)	*p-*Value
Total HEI score				
1st trimester	1.01 [0.99–1.02]	0.68	1.02 [0.99–1.03]	0.52
2nd trimester	0.98 [0.96–0.98]	0.03 *	1.01 [0.98–1.03]	0.33
3rd trimester	0.99 [0.97–1.01]	0.60	1.04 [1.01–1.06]	0.01 *
Interaction term ^1^				
Pre-pregnancy BMI x total HEI score (1st trimester)	1.01 [0.99–1.00]	0.37	1.01 [0.99–1.03]	0.90
Pre-pregnancy BMI x total HEI score (2nd trimester)	0.99 [0.99–1.00]	0.05	1.01 [1.01–1.02]	0.001 **
Pre-pregnancy BMI x total HEI score (3rd trimester)	1.02 [0.99–1.01]	0.51	1.02 [1.01–1.04]	0.001 **

Note. aOR- adjusted odds ratio and 95% CI. Adequate GWG as reference group. Adjusted by age, years of education, parity, physical activity level (MET-hours/week) and pre-pregnancy BMI. ^1^ Only pre-pregnancy BMI showed significant interaction between HEI and GWG. * *p* < 0.05, ** *p* < 0.001.

**Table 5 ijerph-16-03735-t005:** Adjusted OR for associations between total HEI score in each trimester and GWG stratified by pre-pregnancy BMI.

Diet Quality	Inadequate GWG		Excessive GWG	
	aOR [95% CI]	*p*-Value	aOR [95% CI]	*p*-Value
Non-overweight/obese (*n* = 304)				
Total HEI				
1st trimester	1.01 [0.98–1.03]	0.74	1.02 [0.99–1.05]	0.34
2nd trimester	0.97 [0.95–0.99]	0.01 *	0.99 [0.96–1.02]	0.44
3rd trimester	0.99 [0.97–1.01]	0.27	1.04 [1.01–1.07]	0.03 *
Overweight/obese (*n* = 176)				
Total HEI				
1st trimester	1.01 [0.97–1.04]	0.69	0.99 [0.97–1.03]	0.95
2nd trimester	1.00 [0.96–1.03]	0.82	1.04 [1.01–1.07]	0.02 *
3rd trimester	1.01 [0.97–1.05]	0.50	1.04 [1.01–1.08]	0.02 *

Note. aOR- adjusted odds ratio and 95% CI. Adequate GWG was used as reference group. Adjusted by age, years of education, parity, and physical activity level (MET-hours/week). * *p* < 0.05.

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
