# Peer review of "Pre-Pregnancy BMI Influences the Association of Dietary Quality and Gestational Weight Gain: The SECOST Study"

_ijerph, 2019, doi:10.3390/ijerph16193735_

Round 1

Reviewer 1 Report

Comments: This manuscript investigates the role of pre-pregnancy BMI in shaping the association between dietary quality and gestational weight gain. In general, the topic is interesting, but substantial revision is needed, particularly in the empirical analysis and discussion. My specific comments are as follows: First, Table S1 provides the way to calculate HEI, however, I am still not clear about how to measure the score if the real consumption of some food items is greater than the reference. For instance, if the daily consumption of cereals is greater than 8 servings, how much score can I get? By the way, healthy diet does not mean the more the better, overconsumption could also cause a lot of health problem. Second, why the multinomial logistic regression model is used for analyzing the association between HEI and GWG? Women with inadequate GWG, adequate GWG, and excessive GWG are ordered, so I think ordered logistic or linear regression might be better. Third, the explanation of results from multinomial logistic regression (Table 4) is wrong. The odds ratio in the model cannot be explained as compared to women with adequate GWG. It is the change in odds of the specific category (for instance women with inadequate GWG) caused by one unit change in covariates. So that explanation in line 169 should change to “women with higher HEI score has significant lower risk of inadequate GWG”. Fourth, association between HEI and GWG cannot be explained independently since the interaction term is also controlled in the regression. The odds ratio of HEI in table 4 is the one when interaction term equals zero (pre-pregnancy BMI=0). Fifth, discussion in page 9 focuses on dietary consumption at each trimester. However, I do not find any table showing those results. Sixth, I think more work should be done on investigating the role of pre-pregnancy BMI on shaping the association between HEI and GWG. In addition, HEI does not equal dietary quality. It is only one simple indicator to measure dietary quality. I think discussion should be very careful.

Author Response

RESPONSE TO REVIEWERS

Reviewer 1

This manuscript investigates the role of pre-pregnancy BMI in shaping the association between dietary quality and gestational weight gain. In general, the topic is interesting, but substantial revision is needed, particularly in the empirical analysis and discussion.

Response:

Thank you for your valuable comments. We have revised the manuscript accordingly. 

First, Table S1 provides the way to calculate HEI, however, I am still not clear about how to measure the score if the real consumption of some food items is greater than the reference. For instance, if the daily consumption of cereals is greater than 8 servings, how much score can I get? By the way, healthy diet does not mean the more the better, overconsumption could also cause a lot of health problem.

Response:

The score was calculated proportionately between 0 and 10. If an individual consumed less than the recommended amount of servings, the score was calculated with the following formula: 10×(the consumed amount of servings)/(the lower limit of the recommended serving). If an individual consumed more than the recommended amount of servings, the score was calculated with the following formula: 10 – 10×[(the consumed servings)−(the upper limit of the recommended servings)] / (the upper limit of the recommended serving). Each score was rounded off to the nearest whole number. When this calculation produced a negative score because of excess servings, the score was converted to 0 (Line 105-114, text in red).

This formula is based on [1]

Kurotani K, Akter S, Kashino I, Goto A, Mizoue T, Noda M, et al. Quality of diet and mortality among Japanese men and women: Japan Public Health Center based prospective study. BMJ (Online). 2016. Second, why the multinomial logistic regression model is used for analyzing the association between HEI and GWG? Women with inadequate GWG, adequate GWG, and excessive GWG are ordered, so I think ordered logistic or linear regression might be better.

Response:

When women with inadequate GWG (coded 1), adequate GWG (coded 2) and excessive GWG (coded 3) are ordered, the interpretation is based on comparison of the ORs between women with inadequate GWG (reference group) and women with adequate and excessive. However, as inadequate and excessive GWG are both considered as outcomes of unhealthy behaviours (e.g. dietary intake, physical activity) during pregnancy, it would be more meaningful to compare women with adequate GWG (the reference group, coded 3) to women with inadequate (coded 1) and excessive GWG (coded 2). Thus, we would like to retain the present analysis of multinomial logistic regression.

Third, the explanation of results from multinomial logistic regression (Table 4) is wrong. The odds ratio in the model cannot be explained as compared to women with adequate GWG. It is the change in odds of the specific category (for instance women with inadequate GWG) caused by one unit change in covariates. So that explanation in line 169 should change to “women with higher HEI score has significant lower risk of inadequate GWG”.

Response:

We have revised the sentence accordingly. (Lines 179 – 185, text in red)

Fourth, association between HEI and GWG cannot be explained independently since the interaction term is also controlled in the regression. The odds ratio of HEI in table 4 is the one when interaction term equals zero (pre-pregnancy BMI=0).

Response:

The results in Table 4 have been revised accordingly. (line 179 – 185, text in red)

Fifth, discussion in page 9 focuses on dietary consumption at each trimester. However, I do not find any table showing those results.

Response:

Please refer Table 3 on component score and total score for HEI by trimester of pregnancy.

Sixth, I think more work should be done on investigating the role of pre-pregnancy BMI on shaping the association between HEI and GWG. In addition, HEI does not equal dietary quality. It is only one simple indicator to measure dietary quality. I think discussion should be very careful.

Response:

Table 5 shows the adjusted OR for associations between total HEI score in each trimester and GWG stratified by pre-pregnancy to further investigate the differences in HEI and GWG between non-overweight and overweight/obese. We agreed that HEI does not equal dietary quality. The phrase “dietary quality” has been changed to “HEI score”. A paragraph in discussion has been added to highlight the use of HEI to assess diet quality. (Line 293 – 304, text in red)

Reviewer 2 Report

Overall, this paper is very well done content-wise. The majority of comments are grammatical in nature.

Abstract:

Line 17: Add a period after “development” and start a new sentence with “But”

Lines 23-30: add beta coefficients, odds ratios, and p-values after each reported association

Introduction:

Line 35: Spell out the acronym for GWG with first usage. Put a period after “guideline”, then change the next sentence to read “With the exception of Japan, most Asian countries us the IOM GWG guidelines.”

Line 37: delete the word “an”, change guideline to guidelines, delete the “and” and change “recommended” to “, recommending”

Line 38: delete the word “should”

Line 40: add a comma before “while”

Line 42: changed “showed” to “have shown”

Line 44: delete the word “an”

Line 45: delete the word “the”

Line 57: add a comma before “as”

Line 59: change “had” to “have”

Line 60: add comma after miscarriage

Line 63: the phrase “later disease in the infant” is confusing. Do you mean disease during infancy or later in life?

Line 69: Add a comma before “and”

Line 75: Change “until 2 years old” to either “until two years of age” or “until they were two years old”

Line 78: Can you somehow refer the reader to the questionnaire questions?

Line 84: Add “A” before one and hyphenate one-day

Line 101-102: The way this equation is written is very unclear. Please rephrase.

Line 105: Add a period after 100 and start a new sentence with “A higher score …”

Line 108: Add a period after “enrolment” and add another “l” to make spell the word correctly

Line 110-113: The BMI calculation and categories are very standard. You can probably just say BMI was calculated from height and weight and categorized using WHO cut-offs (citation).

Line 119: add comma before “or”

Line 121: change “and” or “or” and add a comma before it

Line 145: add “were” before “employed”

Line 159: Change “For” to “Among”

Line 163: change “was almost” to “were”

Line 176: add a comma before “and”

Table 2: is it possible to shift the headings so they align over the corresponding column of results?

Line 236: delete the word “also” and relocate it to read “were ALSO more likely”

Line 244: add period after “durian” and start a new sentence with “Milk”

Line 245: It’s a little unclear which women “these” is referring to

Line 257: delete comma before “but”

Line 258: Replace “Such” with “Cross-study”

Line 273: Delete “a”

Line 286: Add period after “Malaysia” and start the next sentence as “Most participants were”

Line 295: replace “was” with “such outcomes were”

Author Response

RESPONSE TO REVIEWERS

Reviewer 2

Overall, this paper is very well-done content-wise. The majority of comments are grammatical in nature.

Response:

Thank you. We have revised all the lines accordingly.

Abstract:

Line 17: Add a period after “development” and start a new sentence with “But”

Response:

Revised. (Line 17)

Lines 23-30: add beta coefficients, odds ratios, and p-values after each reported association

Response:

Revised. (Line 23 – 29)

Introduction:

Line 35: Spell out the acronym for GWG with first usage. Put a period after “guideline”, then change the next sentence to read “With the exception of Japan, most Asian countries us the IOM GWG guidelines.”

Response:

Revised. (Line 37)

Line 37: delete the word “an”, change guideline to guidelines, delete the “and” and change “recommended” to “, recommending”

Response:

Revised. (Line 40)

Line 38: delete the word “should”

Responses:

We would like to retain ‘should’ as this is aligned to the recommendation as stated in the preceding statement.

Line 40: add a comma before “while”

Response:

Revised. (Line 43)

Line 42: changed “showed” to “have shown”

Response:

Revised. (Line 44)

Line 44: delete the word “an”

Response:

Revised. (Line 46)

Line 45: delete the word “the”

Response:

Revised. (Line 48)

Line 57: add a comma before “as”

Response:

Revised. (Line 59)

Line 59: change “had” to “have”               

Response:

Revised. (Line 62)

Line 60: add comma after miscarriage

Response:

Revised. (Line 63)

Line 63: the phrase “later disease in the infant” is confusing. Do you mean disease during infancy or later in life?

Response:

Revised. (Line 66)

Line 69: Add a comma before “and”

Response:

Revised. (Line 71)

Line 75: Change “until 2 years old” to either “until two years of age” or “until they were two years old”

Response:

Revised. (Line 78)

Line 78: Can you somehow refer the reader to the questionnaire questions?

Response: Details of the instrument have been published elsewhere [30]. (Line 82)

Line 84: Add “A” before one and hyphenate one-day

Response:

Revised. (Line 88)

Line 101-102: The way this equation is written is very unclear. Please rephrase.

Response:

Revised. (Line 106 – 114)

Line 105: Add a period after 100 and start a new sentence with “A higher score …”

Response:

Revised. (Line 115)

Line 108: Add a period after “enrolment” and add another “l” to make spell the word correctly

Response:

Revised.

Line 110-113: The BMI calculation and categories are very standard. You can probably just say BMI was calculated from height and weight and categorized using WHO cut-offs (citation).

Response:

Revised. (Line 122 – 124)

Line 119: add comma before “or”

Response:

Revised. (Line 130)

Line 121: change “and” or “or” and add a comma before it

Response:

Revised. (Line 132)

Line 145: add “were” before “employed”

Response:

Revised. (Line 156)

Line 159: Change “For” to “Among”

Response:

Revised. (Line 170)

Line 163: change “was almost” to “were”

Response:

Revised. (Line 174)

Line 176: add a comma before “and”

Response:

Revised. (Line 179 – 185)

Table 2: is it possible to shift the headings so they align over the corresponding column of results?

Response:

Revised.

Line 236: delete the word “also” and relocate it to read “were ALSO more likely”

Response:

Revised. (Line 246)

Line 244: add period after “durian” and start a new sentence with “Milk”

Response:

Revised. (Line 253)

Line 245: It’s a little unclear which women “these” is referring to

Response:

Revised. (Line 265)

Line 257: delete comma before “but”

Response:

Revised. (Line 266)

Line 258: Replace “Such” with “Cross-study”

Response:

Revised. (Line 267)

Line 273: Delete “a”

Response:

Revised. (Line 282)

Line 286: Add period after “Malaysia” and start the next sentence as “Most participants were”

Response:

Revised. (Line 306)

Line 295: replace “was” with “such outcomes were”

Response:

Revised. (Line 315)

Round 2

Reviewer 1 Report

Authors have well addressed all of my concerns. I have no further questions.